# Pathophysiological Features of Nigral Dopaminergic Neurons in Animal Models of Parkinson’s Disease

**DOI:** 10.3390/ijms23094508

**Published:** 2022-04-19

**Authors:** Ezia Guatteo, Nicola Berretta, Vincenzo Monda, Ada Ledonne, Nicola Biagio Mercuri

**Affiliations:** 1Department of Motor Science and Wellness, University of Naples Parthenope, 80133 Naples, Italy; ezia.guatteo@uniparthenope.it (E.G.); vincenzo.monda@uniparthenope.it (V.M.); 2Experimental Neurology Laboratory, IRCCS Santa Lucia Foundation, 00143 Rome, Italy; n.berretta@hsantalucia.it; 3Department of Systems Medicine, University of Rome Tor Vergata, 00143 Rome, Italy

**Keywords:** PD toxicity, electrophysiological modifications, firing, excitability, substantia nigra, dopamine, neurotoxin, α-synuclein

## Abstract

The degeneration of nigral dopaminergic neurons is considered the hallmark of Parkinson’s disease (PD), and it is triggered by different factors, including mitochondrial dysfunction, Lewy body accumulation, neuroinflammation, excitotoxicity and metal accumulation. Despite the extensive literature devoted to unravelling the signalling pathways involved in neuronal degeneration, little is known about the functional impairments occurring in these cells during illness progression. Of course, it is not possible to obtain direct information on the properties of the dopaminergic cells in patients. However, several data are available in the literature reporting changes in the function of these cells in PD animal models. In the present manuscript, we focus on dopaminergic neuron functional properties and summarize shared or peculiar features of neuronal dysfunction in different PD animal models at different stages of the disease in an attempt to design a picture of the functional modifications occurring in nigral dopaminergic neurons during disease progression preceding their eventual death.

## 1. Introduction

Parkinson’s disease (PD) is the second most common neurodegenerative disease in humans, characterized by a progressive demise of dopaminergic (DAergic) neurons of the substantia nigra pars compacta (SNpc) [1,2,3], which leads to a dopamine deficit in the dorsal striatum and other target areas [4] responsible for the motor and non-motor symptoms of the disease.

Much of the literature on PD focuses on the morphological, molecular, and inflammatory aspects of the disease based on data obtained from post-mortem specimens of PD patients and animal models. The consistency of the latter was confirmed by the observation that, as in PD patients, they display non-uniform DAergic neuron degeneration, with higher vulnerability in the SNpc, as opposed to the largely spared, at least at initial stages, DAergic neurons of the ventral tegmental area (VTA) [5,6,7,8,9].

Conversely, less attention has been given to defining what functional alteration occurs in the DAergic neuron population preceding their final degeneration due to altered expression or activity of specific ion channels or of the excitatory/inhibitory balance in the network. The importance of this topic should not be underestimated because such modifications may not be just side effects of their progressive deterioration but have a leading role in causing or accelerating the degenerative process as prodromal factors of cellular metabolism impairment. Such a hypothesis can be inferred by simply looking at the higher susceptibility of the DAergic neuron population in the SNpc vs. VTA. Gene-expression studies revealed that the two populations of DAergic neurons share approximately a similar profile [10], indicating that a quantitative rather than a qualitative difference in the expression of a limited number of genes may underlie susceptibility to cell death in PD. 

There is now a high consensus that dendritic, dihydropyridine-sensitive L-type channel-dependent Ca^2+^ waves occurring in SNpc, but not VTA DAergic neurons [11], are possibly involved in the selective SNpc DAergic vulnerability in PD, and currently, a clinical study is ongoing to investigate the potential use of dihydropyridines as neuroprotective PD therapy (reviewed by [12]). Indeed, due to Ca^2+^ accumulation and a highly branched dendritic harbour with an enormous number of DA-releasing sites [13], SNpc DAergic neurons undergo a high level of metabolic stress linked to mitochondrial oxidative phosphorylation to produce ATP necessary for active membrane transportation. Reactive oxygen species (ROS) and superoxide are by-products of oxidative phosphorylation. Thus, a sustained SNpc DAergic neuron biochemical machinery during firing activity may cause constant mitochondrial ROS production. 

The approach we have adopted in this review is to provide an overview of the experimental data obtained from PD animal models on the changes in the functional properties and excitability of nigral DAergic neurons to define standard features of their functional dysregulation during disease progression. 

## 2. Toxin-Based Models of Parkinson’s Disease

Some neurotoxins have been shown to cause DAergic neuron degeneration with a high degree of selectivity, as they enter the neurons via dopamine transporter (DAT; reviewed by [14]), mainly by interfering with the mitochondrial respiratory chain. Taking advantage of this property, these molecules have been extensively used to obtain PD experimental models [15]. These include 6-OHDA, MPTP (MPP^+^), the pesticides rotenone and paraquat, and the non-protein aminoacid L-BMAA.

### 2.1. 6-OHDA

Notably, 6-OHDA was the first agent discovered with specific toxicity towards catecholaminergic neurons [16,17]; thus, it causes selective degeneration of SNpc DAergic neurons and Parkinsonian motor impairments when injected within the substantia nigra, medial forebrain bundle (MFB), or striatum [18,19]. The discovery that 6-OHDA is a naturally occurring compound in the human brain by dopamine hydroxylation [20] and it has been found in PD patients’ urine [21] strengthens the importance of understanding the mechanisms of 6-OHDA toxicity. Thus, 6-OHDA may represent an endogenous toxin contributing to the neurodegeneration of SNpc DAergic neurons in the human brain. When used to produce a PD-like phenotype in an animal model, 6-OHDA-induced death of DAergic neurons is fast, but in some cases, it continues for months after a single toxin brain infusion [22]. The modifications induced by 6-OHDA in surviving SNpc DAergic neurons reported in the literature are pretty scattered, as they were obtained in different experimental conditions regarding the site of toxin injection (SNpc, MFB, striatum), animal species (guinea pig, rat, mouse) and experimental approach (in vivo, ex vivo brain slices, cultured DAergic neurons). 

Our group and others contributed to understanding 6-OHDA toxicity with experiments consisting of 6-OHDA acutely applied onto SNpc DAergic neurons in midbrain slices. The effects of 6-OHDA, evaluated using electrophysiological recordings combined with Ca^2+^ imaging, were remarkably fast and essentially permanent up to 30 min after removal of the toxin, and consisted of a reduction of spontaneous firing activity, partly due to K_ATP_ and D_2_ receptor-gated G-protein-coupled inward rectifier K^+^ channel (GIRK) activation, and a slow-onset Ca^2+^ accumulation, most likely of mitochondrial origin, as it was independent of the removal of calcium ions from the extracellular milieu and emptying endoplasmic reticulum [23]. Similar results were obtained by Qu and colleagues [24]. Another investigation in organotypic slice culture reported that exposure of SNpc DAergic neurons to 6-OHDA for 12–18 h induced changes in their firing pattern from pacemaking to irregular bursting due to enhanced Ca^2+^ influx and increased level and activity of a critical Ca^2+^ modulator protein phosphatase 2A (PP2A), known to alter the Ca^2+^-sensitivity of small-conductance Ca^2+^-activated K^+^ channels (SK) (increased AHP) by dephosphorylating SK-associated calmodulin (CaM). The increased activity of the SK channels is thought to have a protective effect because it reduces cellular damage [25]. Different results were reported when the electrophysiological properties of SNpc DAergic neurons were investigated in brain tissue (midbrain slices or single DAergic neurons) obtained from partially lesioned animals with 6-OHDA injection in vivo. Electrophysiological recordings in isolated TH^+^ neurons performed 1–8 weeks after in vivo 6-OHDA lesion (into the SNpc) in mice revealed that SNpc DAergic neurons in the lesioned side do not undergo maturational changes of resting membrane potential, membrane resistance, steady-state membrane currents, and action potential half-width from 4 to 14 weeks of mouse age, typical of non-lesioned DAergic neurons. Additionally, 6-OHDA SNpc DAergic neurons displayed reduced membrane capacitance, which is indicative of neuron atrophy induced by the toxin. In this lesioned mouse model, the main 6-OHDA effect, other than affecting the maturation of electrical membrane properties of SNpc DAergic neurons, was inhibition of TH expression resulting in an increased proportion of TH^−^ vs. TH^+^ neurons [26].

Other studies investigated 6-OHDA effects on SNpc DAergic neuron electrophysiological properties in vivo, utilizing extracellular single-unit recordings. Four weeks after hemilateral 6-OHDA infusion into the MFB to produce a partial lesion in rats, the remaining SNpc DAergic neurons in the injected side (about a 40% TH^+^ neuron loss) displayed a clear-cut shift toward hyperexcitability, with an increase in firing rate, number of bursts, the mean number of single spikes/bursts and of the percentage of burst-firing neurons [27]. In this report, hyperexcitability is reversed by inhibitors of metabotropic glutamate receptors (mGluRs), suggesting that excessive glutamate release contributes to 6-OHDA toxicity. Another study performed in vivo electrophysiology on SNpc DAergic neurons 2 weeks after hemilateral MFB 6-OHDA lesion and reported a reduced number of spontaneously firing neurons. Still, the remaining active neurons fired at a similar frequency to that of non-lesioned animals [28]. However, the coefficient of variation (CV), a measure of interspike interval (ISI) regularity, was significantly increased in the surviving DAergic neurons, suggesting that 6-OHDA partial lesions result in changes in neuronal excitability, with effects that are limited to the regularity of the firing, which may precede more pronounced hyperexcitability affecting the firing rate at advanced stages of the disease with large striatal dopamine denervation. Recently, one study performed either in vivo single-unit extracellular or ex vivo cell-attached recordings in acute midbrain slices from 6-OHDA lesioned rats (hemilateral MFB, [7]). These authors reported that, in lesioned animals, SNpc DAergic neurons in vivo display a 76% decrease in firing frequency. Interestingly, the decreased firing was restored by GABA_A_ receptor antagonists or MAO-B inhibitors, and it was suggested that reactive astrocytes synthesize and release a high amount of GABA in PD models, which inhibits SNpc DAergic neuron firing (see below in the MPTP section). 

Collectively, 6-OHDA affects several channels/receptors in SNpc DAergic neurons, including K_ATP_ and D_2_-GIRK channel activation, increasing the gating of SK channels, preventing maturation of membrane input resistance, and promoting an excessive release of glutamate and GABA. These effects result in changes in neuronal excitability, disturbance of intracellular ion homeostasis, neuronal atrophy, and mitochondrial dysfunction, as summarized in Table 1.

### 2.2. Rotenone, Paraquat and BMAA

Due to its lipophilic nature, rotenone, a naturally occurring isoflavonoid from tropical plants [29], is membrane permeable and capable of entering all neuron types, where it inhibits complex I of the mitochondria respiratory chain [30]. Despite being an unselective compound, chronic systemic exposure to rotenone in animals has been shown to reproduce some of the key features of PD, including selective degeneration of TH^+^, DAT^+^ and VMAT2^+^ neurons and the formation of α-synuclein-containing intracellular inclusions in nigral DAergic cells [31]. The acute effects of rotenone on SNpc DAergic neurons in midbrain slices have been investigated by numerous groups. One of the main functional impairments caused by the toxin is rapid inhibition of the spontaneous firing and membrane potential hyperpolarization [32,33,34,35]. These effects are associated with mitochondrial depolarization and ROS production, a decrease in membrane input resistance, indicative of K_ATP_ and transient receptor potential M2 (TRPM2) channels opening, and the accumulation of Ca^2+^ and Na^+^ ions. Additionally, the toxin causes a fast drop in cell capacitance, indicative of damage and decline of the cell surface area [34]. With regards to the mechanisms of K_ATP_ channel activation, other studies revealed that rotenone-induced K_ATP_ channel opening is mainly dependent on ROS production rather than on ATP drop [36] since a ROS scavenger (superoxide dismutase mimetic) prevented rotenone-induced K_ATP_ channel activation and membrane hyperpolarization of SNpc DAergic neurons. Interestingly, not all SNpc DAergic neurons display the same sensitivity to rotenone. This feature is due to the diverse composition of K_ATP_ channel subunits in different SNpc DAergic neuron populations, with some K_ATP_ channels demonstrating high (SUR1+Kir6.2) or low (SUR2B+Kir6.2) sensitivity to metabolic inhibition by rotenone [33]. Rotenone also induces detrimental effects on SNpc DAergic neuron excitability via excitotoxic pathways. These include the potentiation of NMDA receptor-mediated currents [37] and inhibition of GABA_A_ receptor-mediated currents [38]. The rotenone effect on NMDA receptors is indirect, mediated by a protein tyrosine kinase-dependent mechanism [37], and it reduces the ability of Mg^2+^ ions to inhibit NMDA-gated channels [39]. The potentiation of NMDA currents by rotenone is also mediated by ROS and/or dopamine oxidation products acting on NMDA receptors indirectly via a protein tyrosine kinase-dependent mechanism [40], indicating that rotenone neurotoxicity may be augmented by dopamine oxidative metabolism. 

Rotenone’s effects on SNpc DAergic neuron excitability have also been studied following chronic systemic administration in vivo to rodents and invertebrates at different toxin concentrations and treatment durations. Intraperitoneal administration of rotenone (0.8 mg/kg) for 7 days to mice does not affect the intrinsic excitability of SNpc DAergic neurons or D_2_ receptor-activated hyperpolarization by exogenous DA application [41]. In this model, rotenone caused impairment of striatal LTP and LTD only in animals with a genetic predisposition to PD (heterozygote PINK1^+/−^ mice) [41]. Another study showed chronic effects of rotenone exposure on the DAergic system in the snail *Lymnaea stagnalis* (0.5 µM, for 7 days). In this model, the hyperpolarizing DAergic response evoked by stimulation of giant DAergic neurons onto post-synaptic VD4 neurons disappeared, indicating that chronic exposure to the toxin impairs DA synaptic transmission [42].

In contrast with the large number of studies aiming at characterizing the electrophysiological effects of rotenone on SNpc DAergic neurons, only a few studies have described paraquat’s electrophysiological effects on these neurons. To our knowledge, the study published by Lee and colleagues [43] is the only one performing an electrophysiological analysis of paraquat effects on SNpc DAergic neurons, showing that acute application of the toxin to rat midbrain slices reduced AMPA-mediated currents by acting selectively on post-synaptic AMPA receptors. Indeed, miniature post-synaptic AMPA current amplitude, but not frequency, was reduced by paraquat as well as the amplitude of AMPA-mediated currents by exogenous agonist application. 

Finally, we need to mention the toxic effects of the non-protein amino acid β-*N*-methylamino-L-alanine (BMAA). Initially, BMAA was proposed as the Cycad toxic agent causing a rare form of neuronal degeneration, the amyotrophic lateral sclerosis–Parkinson’s dementia complex (ALS-PDC), occurring among the Chamorro people of Guam [44,45,46]. However, BMAA is present globally and is produced by cyanobacteria and possibly by other organisms [47,48]. Chronic exposure of non-human primates to BMAA recapitulates the neuropathology as that described in Guamanian people affected by ALS-PDC [49]. BMAA increased the excitability of SNpc DAergic neurons [50] by selectively gating mGluRs, increasing neuronal firing, Ca^2+^ accumulation, and mitochondria ROS production [50]. The main rotenone effects on SNpc DAergic neurons are summarized in Table 2.

### 2.3. MPTP and MPP^+^

The discovery of MPTP and its active metabolite MPP^+^ as a selective toxin of DAergic neurons was made by a neurologist [51] who diagnosed an advanced PD in a young ‘synthetic heroin’ user who accidentally consumed MPTP, a compound previously synthesized (1947) but never controlled for toxicity nor commercialized. Since that discovery, thousands of publications have been made to unveil MPTP’s mechanism of toxicity and gain a translational impact from MPTP animal models in proposing an environmental origin of PD. Indeed, although MPTP has never been found in nature, the structural similarity between its active metabolite MPP^+^, formed by astrocytic MAO, and paraquat, an herbicide largely used worldwide, strongly suggested an increased risk for developing PD following herbicide/pesticide exposure. Primates are highly sensitive to MPTP as well as humans, whereas rodents initially displayed little toxicity to MPTP. This has been explained by the discovery that rats express high levels of MAO enzymes within the blood-brain barrier (BBB), where virtually all lipophilic MPTP is converted into MPP^+^ that conversely, due to its non-lipophilic moiety, does not penetrate the brain [52,53]. Mice display intermediate levels of BBB-MAO and, as expected, intermediate levels of MPTP toxicity. Regarding its effects on the electrophysiological properties of SNpc DAergic neurons, acute MPP^+^ application (10 µM, 5 min) to mouse midbrain slice causes inhibition and subsequent complete cessation of spontaneous firing of these cells [36]. This inhibition is rather selective for SNpc DAergic neurons, as mesolimbic DAergic neurons are largely unaffected by MPP^+^, indicating higher sensitivity of mesostriatal DAergic neurons to the toxin in mice, similar to previous studies on primates [54]. The MPP^+^-mediated hyperpolarization is due to the opening of K_ATP_ channels containing the Kir6.2 subunit, since the spontaneous firing is not altered by MPP^+^ in Kir6.2 knockout mice. The effects of chronic systemic MPTP administration to mice on SNpc DAergic neuron excitability have been recently investigated by Heo and colleagues [7] either in in vivo or in ex vivo midbrain slices from lesioned animals. Four MPTP injections in mice cause a strong (60%) reduction of the spontaneous pacemaker firing as observed in extracellular, single-unit in vivo recordings. The reduction of firing is significantly rescued by treatment of mice with the MAO-B inhibitor selegiline. Indeed, the authors find that following in vivo MPTP (or 6-OHDA) administration, astrocytes become dramatically reactive, as revealed by increased GFAP expression, and begin to synthesize (via MAO-B) and release GABA, with subsequent reduction of SNpc DAergic neuron firing. In line with their hypothesis, ex vivo patch-clamp recordings from SNpc DAergic neurons in midbrain slices confirmed lower pacemaker firing in MPTP slices as in in vivo recordings, and firing reduction is restored by selegiline or the GABA_A_ receptor antagonist bicuculline. The authors conclude that astrocytic GABA release inhibits the pacemaker firing of DAergic neurons in PD models [7]. In line with this evidence, Masi and colleagues [55] similarly reported that MPP^+^ reduces the spontaneous firing of SNpc DAergic neurons when acutely applied to midbrain slices. In addition, they show that MPP^+^ directly inhibits the hyperpolarization-activated current (I_h_), a hallmark of DAergic neurons [56] highly sensitive to pathological conditions of the DAergic system [8,57]. This effect produces an increase in the temporal summation of excitatory inputs to SNpc DAergic neurons, thus increasing spike probability and overall network excitability. Another study described the early and late effects of MPP^+^ on SNpc DAergic neuron excitability [58], confirming the MPP^+^-mediated I_h_ inhibition in Masi et al. [55]. However, this was not causative of firing inhibition/membrane potential hyperpolarization, as it was not prevented by I_h_ blockade. The authors reported that the early MPP^+^ effect was due to DA vesicle displacement and somatodendritic D_2_ receptor activation, whereas late effects depended on K_ATP_ channel activation. MPTP/MPP^+^ effects on SNpc DAergic neurons are summarized in Table 3.

Although not considered a neurotoxin, L-DOPA, the ‘gold standard’ therapeutic drug still largely used to relieve motor symptoms of PD, has been described to have potentially detrimental effects on different types of neurons, including SNpc DAergic cells. Indeed, when the classical inhibitory effect due to neoformed DA has been blocked, L-DOPA displays a non-conventional excitatory effect linked to its nonenzymatic autoxidation to TOPA quinone, a potent activator of AMPA/kainate receptors [59,60,61]. Thus, in addition to the autoreceptor-mediated inhibitory effect of L-DOPA, the drug has the potential to excite and increase free intracellular calcium in nigral DAergic neurons, as well as adjacent non-DAergic cells. This particular circumstance could contribute to neuronal damage.

Overall, in most toxin-based PD models, SNpc DAergic neuron excitability is heavily modified. Acute toxin applications to diverse SNpc DAergic neuron preparations mainly reduce/abolish spontaneous firing, either by gating K_ATP_ channels or increasing GABA release into the midbrain, while in other models, a decrease in firing regularity and potentiation of NMDA_R_-dependent currents is reported. In in vivo toxin-lesioned animals, either an increase or decrease in spontaneous firing has been reported, depending on experimental conditions. These contrasting results suggest that SNpc DAergic neurons’ excitability is a highly sensitive parameter often irreversibly modified by most PD toxins. This has great importance for neuronal health since the increased firing activity of the DAergic neurons drives an increase in intracellular calcium via the L-type channels and causes mitochondrial stress favoring synuclein aggregation [62].

## 3. α-Synuclein-Induced Functional Alterations in Nigral DAergic Neurons

Alpha-synuclein (α-syn) is a small, native intracellular protein encoded by the *Snca* gene located on chromosome 4 in humans [63,64]. It is mainly localized at presynaptic sites in several neurotransmitter systems, where it participates in the regulation of exocytosis by influencing synaptic vesicle trafficking and recycling [64,65,66,67,68], besides affecting neurotransmitter uptake by regulating membrane transporters (DAT; [69]). In addition to the presynaptic compartment, α-syn can also localize to the mitochondrion and nucleus [70,71], wherein it can affect endoplasmatic reticulum (ER)-mitochondrial communication [72].

In physiological conditions, α-syn is an unfolded protein that exists in a dynamic balance between monomeric and oligomeric soluble forms (dimers and trimers). By progressive aggregation, α-syn oligomers convert into small protofibrils and longer fibrils, and ultimately result in insoluble Lewy bodies (LBs), representing the main feature of PD pathology [73,74]. Potential acute triggers of accumulation and aggregation of α-syn are overproduction of the protein or failure in the molecular system that cleaves misfolded forms, as well as exposure to pH changes, oxidative stress, and mitochondrial overwork [75]. Additional mechanisms fostering α-syn aggregation are represented by posttranslational covalent modifications stimulating conformational changes in α-syn structure, including tyrosine nitration (Tyr125), phosphorylation [76,77,78] or proteolytic cleavage at the C-terminus [79], as demonstrated by in vitro tests of α-syn fibrillation or analyses of α-syn forms in LBs [76,77]. 

The first evidence related to α-syn-induced functional alterations in nigral DAergic neurons was described in a bacterial artificial chromosome (BAC) transgenic mouse line expressing wild-type α-syn from the complete human *Snca* locus [80]. In this *Snca* overexpressing model (*Snca*^+/+^ mice), nigral DAergic neurons displayed an age-dependent alteration of spontaneous firing activity, which was reduced in aged mice (18–24 months old) but not in young adult animals (3 months old), as demonstrated by in vivo extracellular recordings under urethane-induced anesthesia [80]. Besides this reduced firing activity, α-syn overexpression, even at the advanced age, has not been associated with other functional alterations in this mouse model, with no variations of the firing regularity (measured as CV-ISI) nor of the proportions of SNpc DAergic neurons displaying various firing modes (i.e., regular, irregular or bursty ones) [80]. Additionally, the movement-related decrease in SNpc DAergic neuron firing seen in in vivo recordings in aged wild-type mice is lost in this *Snca*^+/+^ mouse model of PD, providing evidence that the ‘real-time’ encoding of behavior by the firing of DAergic neurons is perturbed in Parkinsonism [81].

Further evidence of α-syn-induced functional changes in nigral DAergic neurons has been provided in a mouse model overexpressing the mutated A53T form of α-syn (A53T-SNCA mice, [82]). Precisely, utilizing extracellular in vivo recordings under anesthesia, it has been reported that A53T-SNCA overexpression causes an age-dependent increase in the spontaneous firing frequency of nigral DAergic neurons. These neurons recorded from young adult A53T-SNCA mice (3–4 months old) displayed a milder increase in spontaneous firing frequency, associated with regularity of pacemaker firing (reduction of the CV-ISI). This increased spontaneous firing rate became more prominent in older A53T-SNCA mice (7–10 months old), but without variation in the firing regularity, and was not associated with an increased trend to burstiness (measured as the percentage of spikes fired as bursts). A53T-SNCA overexpression, at advanced stages, affects the action potential waveform of nigral DAergic neurons, with a prolongation of the repolarization phase [82]. An impairment of voltage-activated K^+^ channels (subfamily Kv4.3) seems to underlie the potentiation of firing rate, as it was occluded by pharmacological inhibition of Kv4.3 channels. Remarkably, functional alterations induced by A53T-SNCA overexpression were restricted to DAergic neurons of SNpc, whereas those in the adjacent VTA also displayed similar spontaneous firing activity in the advanced age in comparison with age-matched controls [82]. 

Alpha-synuclein-induced functional alterations of nigral DAergic neurons has been analyzed in another PD rat model in which α-syn is spontaneously overexpressed as a consequence of a point mutation in the 3′ untranslated region of *Snca* mRNA that increases transcription of the protein [8,83]. Such mutated rats, during the first weeks of age, displayed an increase in α-syn protein expression levels in several brain regions, including the mesencephalic area, and early signs of degeneration of DAergic neurons in SNpc, but not in VTA [8]. Notwithstanding, functional properties of surviving nigral DAergic neurons were mostly preserved, with no overt alterations in the spontaneous firing nor in the intrinsic excitability in mutated rats [8]. Interestingly, spontaneous α-syn overexpression was associated with a decrease in the I_h_ current, mediated by the HCN channels [8], an electrophysiological signature of nigral DAergic neurons, that is similarly reduced in the MPTP PD model [55,84], whereas the function of the DA D_2_ autoreceptor and GABAergic GABA_B_ were not affected [8].

Additional insights on the pathological alterations of nigral DAergic neurons reliant on α-syn overexpression have been provided by electrophysiological investigations in a bacterial artificial chromosome (BAC) transgenic rat model of PD, which overexpressed the full-length human α-syn (*Snca*^+/+^ rats) [85,86]. The functional changes of nigral DAergic neurons in this α-syn-overexpressing rat model (overt in 4-month-old *Snca*^+/+^ mice, but not in younger 2-month-old animals) consisted of a reduction in spontaneous firing frequency, along with a reduction in the firing regularity, and a decrease in intrinsic excitability, as demonstrated by patch-clamp whole-cell recordings in ex vivo midbrain slices from *Snca*^+/+^ in comparison with control rats [86]. Moreover, the afterhyperpolarization-activated current (I_AHP_)—an intrinsic current that underlies the afterhyperpolarization phase of each action potential—was increased in adult *Snca*^+/+^ rats, thus supporting that α-syn overexpression, by prolonging the AP afterhyperpolarization phase, influences the firing frequency of nigral DAergic neurons [86].

A recent investigation clarified the functional effects of the accumulation of α-syn aggregates on the activity of nigral DAergic neurons [87]. It has been demonstrated that the intrastriatal injection of α-syn-preformed fibrils (α-syn-PFF) in rats, which retrogradely accumulate in nigral DAergic neurons, perturbed their spontaneous firing activity in a bidirectional and time-dependent manner [87]. Precisely, in rats subjected to intrastriatal α-syn-PFF injection, the spontaneous firing rate of nigral DAergic neurons was inhibited at an early stage (6 weeks after injections), whereas at a later time-point (12 weeks after injection), these neurons displayed increased firing frequency and enhanced excitability [87]. Such changes in the firing activity were not associated with other functional changes which might affect action potential generation, frequency, and regularity, such as modifications in the I_h_ current or dysfunctions in the activity of D_2_ receptors [87], that typically provide a DA-mediated autoinhibition of nigral DAergic neurons in physiological conditions [88]. 

Acute functional effects of α-syn aggregates (oligomeric forms, plus a small proportion of fibrils) on SNpc DAergic neurons have been recently described by analyzing the electrophysiological modifications using the whole-cell patch-clamp technique [89]. Intracellular injections of α-syn aggregates, but not α-syn monomeric forms, progressively reduced membrane resistance and decreased the firing rate and excitability of nigral DAergic neurons [89]. A contribution of ATP-sensitive K^+^ channels (K_ATP_)—a family of channels that activates during energy depletion—has been suggested as the underlying mechanism of such α-syn-induced reduction of overall activity/excitability of DAergic neurons, based on the evidence that α-syn’s effects were partially counteracted in the presence of the K_ATP_ inhibitor glibenclamide [89]. 

Altogether, current evidence regarding α-syn-induced effects on the function of DAergic neurons has provided insights into time-dependent modifications that mainly affect spontaneous firing activity and excitability of these neurons in a bidirectional way. Acute α-syn-induced effects (as intracellular injection of α-syn oligomers; [89]), or early time-points in α-syn accumulation/aggregation, as resembled 1) in α-syn overexpressing animals at a young age [86], 2) when α-syn overload is still preceding overt α-syn aggregation [80], 3) during initial phases of α-syn-PFF accumulation [87], are associated with inhibition of firing activity of DAergic neurons. Contrariwise, increased spontaneous firing activity and excitability of these cells are observed along advanced stages of α-syn aggregate overload [87] or following the overexpression of mutated α-syn forms (as in A53T-SNCA), which might foster α-syn pathological aggregation and pathology [82]. 

Therefore, mechanisms governing spontaneous firing frequency and excitability in nigral DAergic neurons are preferential targets of vulnerability during the progression of α-syn overload. Various intrinsic factors, in addition to extrinsic/synaptic inputs, are required to generate and homeostatically tune the activity patterns of these cells to physiological demands. Future research needs to clarify the precise molecular events underlying α-syn-induced functional alterations. Some studies reported modifications in the intrinsic I_h_ current [8] and I_AHP_ [86], impairment of voltage-dependent K^+^ channels [82] (all involved in the control of action potentials rate and fidelity), or increased K_ATP_ activation (which fosters neuronal hyperpolarization) [89]. Nevertheless, I_h_ is preserved in nigral DAergic neurons from other models of α-syn-based PD animal models [86,87]. Thus, additional studies are required to confirm the precise molecular substrates of dysfunctions caused by α-syn aggregation/accumulation on nigral DAergic neurons. The effects of α-syn on SNpc DAergic neurons are summarized in Table 4.

## 4. Other Genetic Models

In addition to genetic models carrying mutations targeting α-syn function, several rodent models have been developed, characterized by mutations associated with familial forms of PD. Among these, the most commonly used mutations include those in genes encoding for PTEN-induced putative kinase 1 (PINK1), parkin, leucine-rich repeat kinase 2 (LRRK2), and protein deglycase DJ-1 (DJ-1) or targeting mitochondrial function (MitoPark).

### 4.1. PINK1 and Parkin

PINK1 is a serine/threonine kinase present in the mitochondria, where it cooperates with ubiquitin E3 ligase parkin in favoring the removal of damaged organelles, thereby exerting a protective action [90,91,92]. Mutations in the autosomal recessive PINK1 and Parkin genes have been associated with the PARK6 and PARK2 variants of early-onset PD [93,94]. Accordingly, PINK1- and parkin-mutated mice models have been developed, displaying motor impairments associated with reduced DA release in the striatum and mitochondrial dysfunctions [95,96]; see [97] for review. With regards to functional modifications of the SNpc DAergic neuronal populations, Bishop and colleagues [98] described an increased irregularity of the SNpc DAergic neuron firing in mature animals (3–4 months) in PINK1-deficient mice, measured both in vivo and ex vivo, whose ionic mechanism can be ascribed to a reduced I_AHP_ due to opening of SK channels. In addition, the same authors indicate a reduced release of Ca^2+^ from intracellular stores, including the endoplasmic reticulum and mitochondria, as possible causes underlying the reduced activation of the Ca^2+^-dependent SK currents.

An increased firing rate of SNpc DAergic neurons recorded ex vivo has also been observed in ParkinQ311X mice expressing a human variant of Parkin [99]. However, in this case, the mechanism underlying the increased excitability is shown to reside in a higher expression of kainate glutamate receptors (KAR), because the pharmacological antagonism of these receptors restores the firing to the control rate and prevents neurodegeneration of SNpc DAergic cells. A subsequent study conducted in vivo on the same animal model by the same group reported modifications in the bursting behavior of these neurons rather than in their overall firing rate [100]. Since burst firing of the DAergic neurons in vivo is believed to be linked to the presence of glutamatergic afferents that are largely excised in slice preparations, the authors suggest that the higher KAR expression of ParkinQ311X mice may be responsible for the change in pattern, rather than frequency, of the SNpc DAergic neurons in living animals.

The importance of the glutamatergic drive is also suggested by a report from PINK1-mutated mice, providing evidence that the functional impairments of SNpc DAergic neurons may begin even shortly after birth [101]. These authors compared immature (P2–P10) and young adult (1–3 months) mice among wild-type and PINK1-mutated animals. They found differences in SNpc DAergic neurons’ passive properties expressed at an early postnatal age, although these differences became less evident at later stages of development. More interestingly, they found a reduced presence of spontaneous NMDAR-mediated synaptic currents in immature SNpc DAergic neurons, which is possibly involved in changes in somatodendritic calcium dynamics and, consequently, neuronal maturation. Further investigation utilizing NMDAR subunit-selective pharmacology led the authors to suggest a different profile in the dynamics of specific NMDA subunits in glutamatergic synapses of PINK1-mutated pups.

### 4.2. LRRK2

LRRK2 is a serine-threonine kinase that plays a role in several cellular functions, including endolysosomal vesicle trafficking, mitochondrial turnover, and autophagy. The LRRK2 gene is associated with the PARK8 variant, whose mutation causes an autosomal dominant form of PD, which is responsible for most of the known heritable forms of PD [102], although the precise mechanisms by which LRRK2 mutation causes degeneration of DAergic neurons have not as yet been completely elucidated. Several mutant murine models have been developed to investigate LRRK2’s role in PD, displaying functional, histological, and behavioral alterations resembling a PD-like phenotype (see [103] for review). With regards to alterations of SNpc DAergic neuron function, reduced excitability has been reported in 8-month-old mice carrying the G2019S LRRK2 mutation, expressed as reduced firing without affecting their passive properties [104]. Conversely, a more recent report on mice carrying the same G2019S LRRK2 mutation did not confirm modifications in neuronal firing; rather, it described alterations in the afferent glutamatergic transmission, as indicated by a reduced synaptic glutamate barrage and of presynaptic molecular markers of glutamatergic terminals in the SNpc, as opposed to VTA [105].

A reduction in the firing of SNpc DAergic neurons has also been reported in vivo in bacterial artificial chromosome transgenic rats expressing the R1441C LRRK2 mutation. In this case, the authors report a reduced firing variability due to reduced bursting of the recorded DAergic neurons [106].

### 4.3. DJ-1 and MitoPark

Another gene whose mutation has been linked to PD pathogenesis is DJ-1. In particular, DJ-1 is associated with the PARK7 variant, reported as second in the rank of identified inherited PD [107]. DJ-1 is highly expressed in neurons and glial cells, playing a significant role against oxidative stress and inflammatory processes by altering the mitochondrial electron transport respiratory chain [108], and it is probably because of this protective action that DJ-1 mutation gives rise to detrimental consequences, especially in those cells with higher metabolic demand, such as the SNpc DAergic neurons (see [109] for review).

Although no clear evidence of neurodegenerations exists, a higher vulnerability of the SNpc DAergic neuron population to neurotoxic agents has been reported in DJ-1 knockout models [27,110]. In agreement with this histopathological evidence, no clear functional modifications in their basal excitability have been uncovered; however, the post-synaptic D_2_ receptor-mediated inhibitory response, typical of this neuronal population [88,111], is significantly reduced, indicating an impairment of D_2_ receptor-mediated responses [112]. A subsequent investigation on the same animal model confirmed the D_2_ autoreceptor dysfunctions. Moreover, an enhanced membrane potential response was observed when DAergic neurons from DJ-1 knockout mice were exposed to oxygen and glucose deprivation, rotenone, or the pharmacological block of the Na/K ATPase pump, indicative of a higher susceptibility of these neurons to agents causing metabolic stress [113].

A genetic PD model has been developed that, similarly to DJ-1 mice, targets mitochondrial function, named MitoPark mice. These animals are characterized by a mutation of mitochondrial transcription factor A selectively on the DAergic neurons and display distinct features of a PD-like phenotype, including progressive loss of DAergic neurons in the SNpc, decreased DA level in the striatum, and locomotor impairments, improved by L-DOPA treatment [114,115]. An electrophysiological investigation of the membrane properties of SNpc DAergic neurons has been first performed on MitoPark mice at 6–8 weeks of age in midbrain slices. Despite the pre-symptomatic stage, a significant reduction in HCN-mediated I_h_ current was reported, together with an anomalous diversification in the firing properties of these neurons. In particular, while the large majority of the DAergic neurons displayed their typical tonic spontaneous firing in control conditions, only a small portion of the recorded neurons in MitoPark mice had similar characteristics, while most neurons were silent or fired at an abnormally high rate [116]. A subsequent investigation on the same MitoPark animal model revealed a progressive change in the functional properties of the DAergic neurons from 6–10 to more than 16 weeks of age. Thus, a tendency to decreased membrane capacitance and increased membrane resistance is already present in the early stages, becoming more significant in later stages. Similarly, a progressive reduction in HCN-mediated I_h_ and SK-mediated I_AHP_ can be relieved in MitoPark mice. With regards to DA neurons’ tonic firing, no clear change is reported at all tested ages; however, a reduced CV-ISI develops with age in mutant mice, indicating a trend towards a higher irregularity of the firing [117]. The same authors also investigated DA release of the SNpc DAergic neurons, together with their response to D_2_ autoreceptor stimulation. Notably, they found a significant reduction of the post-synaptic response to DA in SNpc DA neurons of aged MitoPark mice and a reduced release of DA, measured as amphetamine-induced currents and synaptically-evoked DA responses, at both early and later stages [117].

More recently, a novel mutant mouse has been developed that similarly targets mitochondrial function, consisting of conditional knockout mice for the *Ndufs2* subunit of the mitochondrial complex-I of the DAergic neurons. At 30 days postnatal age, when DA release in the striatum was significantly reduced, the authors reported reduced neuronal excitability of the SNpc DAergic neurons, with a lower tonic firing rate and associated intracellular Ca^2+^ dynamics, and a smaller I_h_, although the same neurons tended to respond with a higher frequency burst when subject to glutamate-mediated excitation [118]. The effects of genetic manipulations on SNpc DAergic neurons are summarized in Table 5.

The overall picture emerging from murine genetic models largely confirms that the progression of a PD-like phenotype is not merely linked to morphological and biochemical alterations of the SNpc DAergic neurons but also to functional modifications in their electrical properties preceding and possibly cooperating with those cellular homeostasis impairments that move the DAergic neuron population towards an irreversible neurodegenerative pathway. The direction of the change, either towards hyperexcited or inhibited conditions, may depend on the stage of PD phenotype development; however, a change in the firing rate or the firing pattern, favoring bursting vs. regular firing behavior and vice versa, seems to be a clear hallmark of disease progression. Equally, the ionic mechanisms underlying these modifications are not uniform since both altered intrinsic membrane currents, particularly I_h_ and I_AHP_, and imbalances in the excitatory vs. inhibitory synaptic drive may occur, targeting the glutamatergic excitatory transmission or the typical D_2_ receptor-mediated self-inhibition of the midbrain DAergic neurons.

## 5. Summary of DAergic Neuron Functional Alterations in PD Models

PD is linked to dysfunctions of DAergic neurons located in the ventral midbrain, which result in the reduction of DA release in the projecting areas of the brain. DA release is strongly dependent on the firing pattern of DAergic neurons. In vivo, DAergic neurons display two main types of spontaneous firing patterns (recently reviewed by [119]), single spike firing and burst firing, characterized by single action potential discharge (as a regular or an irregular pattern) and clusters of two-to-ten action potentials followed by single action potentials, respectively [120,121]. Within the SNpc, the vast majority of DAergic neurons fire in a single action potential mode with different degrees of regularity to maintain the DA tone in the target areas, such as the dorsal striatum [122]. The regularity of the single spike firing depends on specific ion channel activity. SK channels, in particular, have been shown to strongly affect the firing regularity in DAergic neurons, measured as CV-ISI. Indeed, both SK channel inhibition by apamin [123,124] and SK channel activation by positive modulators [123] promote an irregular firing in SNpc DAergic neurons accompanied by an increase in their CV. Interestingly for the scope of the present review, a decrease in firing regularity of SNpc DAergic neurons linked to modifications of SK channel activity has been reported in different in vitro and in vivo models of PD [25,86,98], suggesting that it may represent a functional parameter predisposing SNpc DAergic neurons to selective vulnerability in PD. Thus, (a) 18h exposure of DAergic neurons to 6-OHDA increases SK channels activity by increasing intracellular Ca^2+^ concentration [25], (b) neuroinflammation caused by *Snca* overexpression in rats augments SNpc DAergic neurons SK channel-mediated current and reduces firing regularity [86]; (c) a reduction in SK channel activity, reported in PINK1-deficient mice, causes an irregular firing pattern of SNpc DAergic neurons [98].

In α-syn-based PD models, bidirectional modifications of the firing frequency of nigral DAergic neurons have been reported as either a reduction [80,86,87,89] or an increase [82,87]. The latter could be related to the α-syn-dependent progression of PD pathology [87].

The switch from single action potential pacemaker firing to bursting in SNpc DAergic neurons depends on synaptic inputs, mainly glutamate acting on NMDA receptors (reviewed by [119]. In some PD models, the firing irregularity of SNpc DAergic neurons of lesioned animals was also dependent on mGluR activation, since a selective mGluR antagonist restored firing regularity [27]. Other ionic conductances affecting firing rate or regularity/bursting, targeted by PD toxins or genetic manipulations, include the K_ATP_ and I_h_ currents [8,23,24,27,32,33,34,35,36,55,58,89], as summarized in Figure 1.

## 6. Conclusions

The responses of DAergic cells of the substantia nigra to toxic agents, α-syn overload, and genetic manipulations, although diverse, given the different insults and the modality of their expression, could be generally recapitulated as acute and chronic. The bioenergetic failure of the cells due to mitochondrial impairment and the high metabolic demand of DAergic cells during their pacemaker or bursting activity promptly activates the opening of K_ATP_ channels, which brings the cells to a silent state. This phenomenon, by hyperpolarizing the membrane, could initially contribute to limiting sodium and especially calcium overload into these cells, thus reducing mitochondrial reactive oxygen species and oxidative stress (Figure 1). Despite this, because of the progressive, damaging processes, some cells would inevitably die, and those that survive change their firing discharge in an attempt to limit their demise and compensate for DA deficits. These firing modifications are mainly due to the modification of membrane properties (e.g., capacitance and resistance) and ion channel activity. Therefore, different pictures of functional states could be present, being mainly inhibitory at the early PD stage and mainly excitatory at the advanced stage (Figure 1). The early inhibitory responses are thought of as a continued effort of the cells to maintain their metabolic state, while the later excitatory responses could be interpreted as a compensatory strategy of the DAergic neurons aiming at maintaining extracellular DA to a sufficient level to control movement and cognition before their irreversible decline and ultimately death.

In conclusion, greater attention is demanded towards the inclusion of DAergic neuron excitability as an important parameter in studies focused on the cellular mechanisms of PD using animal models in vivo and in vitro, as this may prompt the identification of novel targets for pharmacological interventions that, by firing modulation, could alleviate PD symptoms and eventual progression.

## Figures and Tables

**Figure 1 ijms-23-04508-f001:**
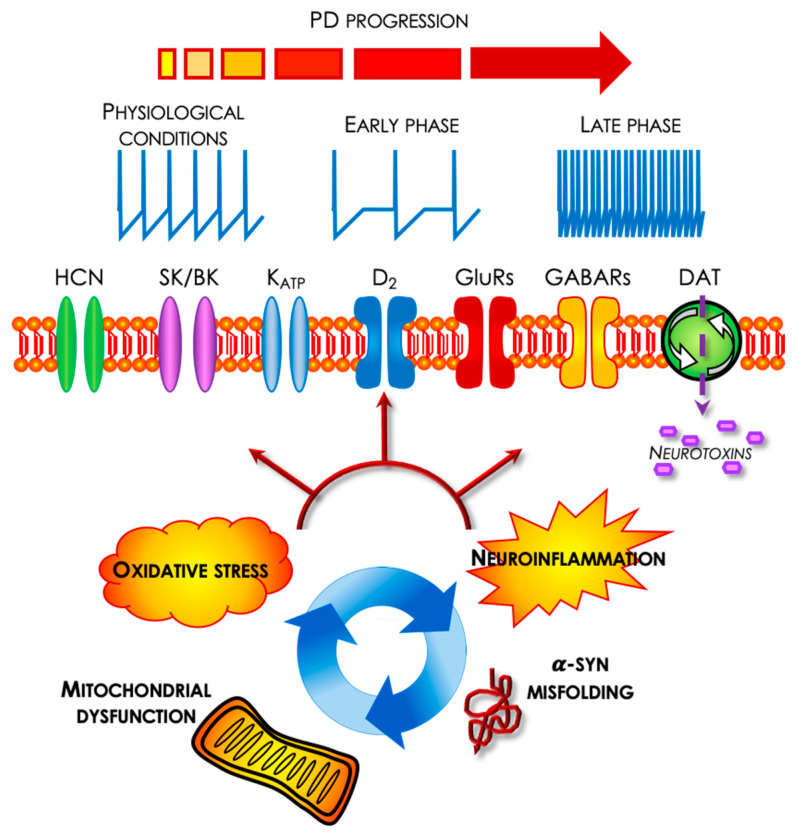
Schematic representation of the main cellular and molecular processes responsible for PD pathogenesis, including neurotoxic agents entering SNpc DAergic neurons through the DA transporter (DAT). During the course of PD progression, functional alterations occur in this neuronal population due to changes in the biophysical properties of several ionic channels, like HCN channels, responsible for I_H_, broad (BK) and small (SK) conductance Ca^2+^-dependent K^+^ channels, responsible for spike afterhyperpolarization (I_AHP_) and ATP-dependent K^+^ channels. Alterations of the synaptic network also contribute to changes in membrane excitability involving glutamate and GABA transmission, as well as local D2 receptor-mediated auto-inhibition. Overall, these membrane mechanisms alter DAergic neurons’ resting membrane potential and firing discharge. Firing inhibition seems to characterize the initial stages of PD progression, possibly as an early defensive response against mitochondrial dysfunction, in order to preserve energy consumption. Conversely, later stages of PD progression appear to be mostly associated with an increase in membrane excitability, possibly in order to compensate for the loss of DA transmission, due to the ongoing neurodegenerative process eventually also affecting these hyperactive SNpc DAergic neurons.

**Table 1 ijms-23-04508-t001:** Summary of the principal findings of the cited literature on 6-OHDA effects on SNpc DAergic neuron functional properties.

Type of Study	[6-OHDA]	Treatment	Modified Parameters in SNpc DAergic Neuron	Molecular Mechanisms	Reference
Ex vivo, rat	0.2, 0.5, 1, 2 (mM)	5 or 10 min	Inhibition of spontaneous firing; R_m_ drop; Ca^2+^ accumulation	D_2_-GIRK and K_ATP_ channels activation; mitochondrial release of Ca^2+^ ions	[23]
Ex vivo, rat	0.5, 1, 2 (mM)	3–5 min	Inhibition of spontaneous firing; Ca^2+^ accumulation	N-type VGCC current amplitude increase	[24]
In vitro organotypic culture, rat	25 µM	12 or 18 h	Irregular firing/bursting; depolarized RMP	Increased AHP and I_AHP_ mediated by SK channels	[25]
In vivo, mouse	1.5 µg/µL (1.6 µL)	1 injection, SNpc	1 to 8 weeks after lesion; Lack of maturation of R_m_, AP half-width, steady-state I_(-100mV)_		[26]
In vivo, rat	4 µg/4 µL	1 injection, MFB; tested 16–20 days after lesion	Increase in firing rate, n. of bursting neurons and n. spikes/burst	Release of glutamate and mGluR activation (rescue by MPEP)	[27]
In vivo, rat	4 µg/2 µL	1 injection, MFB, 4–6 weeks after lesion	Decreased n. of active neurons; no significant difference in firing rate nor bursting; higher CV	Rearrangements of circuitry to compensate for neuronal loss	[28]
In vivo, rat	8 µg/4 µL	1 injection, MFB	32 days after lesion, 76% reduction in firing rate	Excessive GABA release by reactive astrocytes, rescued by MAO inhibitor safinamide	[7]
Ex vivo	Ipsilateral slices from in vivo lesioned rat		Increase tonic GABA_A_ current; no difference in sIPSC amplitude or frequency	Rescued by bicuculline and safinamide	[7]

**Table 2 ijms-23-04508-t002:** Summary of the principal findings of the cited literature on rotenone’s effects on SNpc DAergic neuron functional properties.

Type of Study	[Rotenone]	Treatment	Modified Parameters in SNpc DAergic Neuron	Molecular Mechanisms	Reference
In vitro, dissociated SNpc DAergic neurons, rat	5 µM		Firing inhibition and membrane hyperpolarization	Activation of the sulphonylurea-sensitive K_ATP_ current	[32]
Ex vivo midbrain slice, mouse	10 µM	10 min	Firing inhibition and membrane hyperpolarization	SUR1-Kir6.2 vs. SUR2B-Kir6.2 K_ATP_ channels display different sensitivity to metabolic inhibition	[33]
Ex vivo midbrain slice, rat	5 nM; 200 nM; 1 µM	10 min	C_m_ and R_m_ drop; K_ATP_ current activation; Ca^2+^ and Na^+^ accumulation; mitochondrial ROS production and Δψ_m_ depolarization	ROS activation of TRPM2 Ca^2+^-permeable and K_ATP_ channels	[34]
In vitro, SNpc DAergic neurons acutely dissociated	1 µM	5–6 min	Firing inhibition	K_ATP_ channel opening; they are inhibited by the neuroprotective agent THB	[35]
Ex vivo, midbrain slices, mouse	100 nM	5 min	Firing inhibition; K_ATP_ channel activation; ROS production	Kir6.2 subunit KO prevents DAergic neuron degeneration	[36]
Ex vivo, midbrain slices, rat	100 nM	20–30 min	Increased I_NMDA_ (but not I_AMPA_) amplitude		[37]
In vitro, acutely dissociated SNpc DAergic neurons, rat	5 µM	10 min	Run-down of I_GABAA_, but not of I_Gly_ or I_Glu_		[38]
Ex vivo midbrain slices, rat	100 nM	30 min	Increased I_NMDA_ amplitude	Loss of Mg^2+^-block of NMDA-mediated currents that involves a tyrosine kinase	[39]
Ex vivo midbrain slices, rats	100 nM	30 min	Increased I_NMDA_ amplitude	ROS and DA oxidation products mediate NMDA currents increase	[40]
In vivo, mouse	0.8 mg/kg	7 days	Lack of gross functional alterations in SNpc DAergic neurons		[41]
In vivo, snail Lymnaea stagnalis	0.5 µM	7 days	Loss of dopaminergic IPSP	Uncoupling of dopaminergic synapses	[42]
Ex vivo midbrain slice, rat	Paraquat, 30,100 µM	20 min	Reduced I_AMPA_ amplitude	Inhibition of post-synaptic AMPA receptors	[43]
Ex vivo midbrain slice, rat	BMAA (0.1–10 mM)	2–3 min	Increased firing; Ca^2+^ accumulation	Activation of mGluR and TRPC channels	[50]

**Table 3 ijms-23-04508-t003:** Summary of the principal findings reported by the cited literature on MPP^+^/MPTP effects on SNpc DAergic neuron functional properties.

Type of Study	[MPTP/MPP^+^]	Treatment	Modified Parameters in SNpc DAergic Neuron	Molecular Mechanisms	Reference
Ex vivo	100 nM–10 µM	5 min	Spontaneous firing inhibition; K_ATP_ activation	Differential coupling between mitochondrial inhibition and K_ATP_ activation in SN vs. VTA neurons. Kir6.2 subunit KO prevents DAergic neuron degeneration	[36]
In vivo	20 mg/kg, i.p., 4 injections in one day	6 days later	60% reduction of pacemaker firing	Excessive GABA release by reactive astrocytes	[7]
In vitro, acutely isolated DAergic neurons from in vivo lesioned mouse			Decrease in spontaneous firing rate	Excessive GABA release by reactive astrocytes, (rescue by selegiline and bicuculline);	[7]
Ex vivo midbrain slices, rat and mouse	50 µM	5–15 min	I_h_ inhibition; spontaneous firing inhibition	The shift of I_h_ activation curve toward negative potentials	[55]
Ex vivo	20 µM	30 min	Spontaneous firing inhibition	DA vesicle displacement, D_2_-GIRK activation; I_h_ inhibition; K_ATP_ activation; DAT activation	[58]

**Table 4 ijms-23-04508-t004:** Summary of the principal findings of the cited literature on α-synuclein-dependent effects on SNpc DAergic neuron functional properties.

α-Synuclein-Related Manipulation	Specie	Methodological Information	Age	Modified Parameters in SNpc DAergic Neurons	Molecular Mechanisms	Reference
BAC-induced overexpression of human Snca	Mice (C57/Bl6 background)	In vivo single-unit extracellular recordings in urethane-anesthetized mice	3–4 months 18–22 months	No alterations Reduced spontaneous firing rate		[80]
Overexpression of mutated A53T-Snca	Mice (C57BL6 background)	In vivo single-unit extracellular recordings in urethane-anesthetized mice	3–4 months 7–10 months	- Increased spontaneous firing; - Reduced CV-ISIs; - Increased spontaneous firing frequency; - Increased action potential (AP) repolarization phase.	Age-dependent impairment of voltage-activated K^+^ channels due to redox species	[82]
Spontaneous overexpression of α-syn	Rat	In vivo single-unit extracellular recordings in urethane-anesthetized mice	21–30 days	- No alterations in spontaneous firing; - No alterations in evoked excitability; - Normal D2-activated currents; - Normal GABAB-activated currents; - Reduced I_h_.		[8]
BAC-induced overexpression of human Snca	Rat (backgroud SD)	Ex vivo patch-clamp recordings in horizontal acute midbrain slices	5 months	Decrease in spontaneous and evoked firing; increase in CV	Increase of I_AHP_	[86]
Intrastriatal injection of α-syn-PFF	Wistar Rat	Ex vivo patch-clamp recordings in horizontal acute midbrain slices from 4–6 months-old rats subjected to in vivo intrastriatal α-syn-PFF injections (6 or 12 weeks before recordings)	6 weeks after α-syn-PFF injection	- Increased frequency of spontaneous firing; - Increased intrinsic excitability; - Normal I_h_; - Normal D2-activated K^+^ currents.		[87]
			12 weeks after α-syn-PFF injection	- Increased frequency of spontaneous firing; - Increased intrinsic excitability; - Normal I_h_; - Normal D2-activated K^+^ currents.		
Acute injection of α-syn aggregates (oligomers and small fibrils) in single DAergic neurons	C57/BL6 mice	Ex vivo patch-clamp recordings in coronal acute midbrain slices	2–3 weeks	- Reduction in R_m_; - Reduced spontaneous firing; - Reduced excitability.	α-syn-induced activation of K_ATP_	[89]

**Table 5 ijms-23-04508-t005:** Summary of the principal findings reported by the cited literature on genetic manipulation effects on SNpc DAergic neuron functional properties.

Gene Mutation	Age	Methodological Information	Modified Parameters in Snpc DAergic Neuron	Reference
PINK1	3–4 months	In vitro patch-clamp recordings and in vivo single-unit recordings in urethane-anesthetized animals	- No change in basal firing rate but higher irregularity in their pattern; - Increased burst firing in vivo ; - Reduced I_AHP_ mediated by SK channels; - Impaired Ca^2+^ release from the endoplasmic reticulum and mitochondria	[98]
PINK1	6–7 days 1–3 months	In vitro patch-clamp recordings	- Higher membrane capacitance and Ih current density; - Reduced NMDA-EPSCs; - Higher input resistance; - Reduced NMDA-EPSC	[101]
Parkin	25 days	In vitro cell-attached recordings	- Increased firing rate; - Increased KAR expression	[99]
Parkin	30 days	In vivo single-unit recordings in chloral hydrate-anesthetized animals	- Increased number of spikes within the burst	[100]
LRRK2	8 months	In vitro patch-clamp recordings	- Reduced firing rate	[104]
LRRK2	16–22 months	In vivo single-unit recordings in urethane-anesthetized animals	- Reduced bursting behavior	[106]
LRRK2	10–12 months	In vitro patch-clamp recordings	- Reduced synaptic glutamatergic drive	[105]
DJ-1	1 month	In vitro patch-clamp recordings	- Reduced D_2_ receptor-mediated responses	[112]
DJ-1	1–2 months	In vitro patch-clamp recordings	- Enhanced response to ODG, rotenone and block of the Na/K pump	[113]
Mitochondrial *Tfam* (MitoPark)	6–8 weeks	In vitro patch-clamp recordings	- Reduced I_h_; - Abnormal pacemaker firing (mostly silent neurons or with a high rate)	[116]
Mitochondrial *Tfam* (MitoPark)	6–10 weeks >5 months	In vitro patch-clamp recordings	- Reduced cell capacitance; - Reduced I_h_ and SK-mediated I_AHP_; - Reduced amphetamine- and synaptically evoked DA release; - Reduced cell capacitance and increased input resistance; - Lower regularity in the tonic firing; - -Reduced I_h_ and SK-mediated I_AHP_; - Reduced D_2_ receptor-mediated responses; - Reduced amphetamine- and synaptically evoked DA release	[117]
*Ndufs2* of mitochondrial complex-I	1 month	In vitro patch-clamp recordings	- Reduced tonic firing rate; - Reduced I_h_; - Reduced Ca^2+^ oscillations; - Increased spikes in evoked bursts	[118]

## Data Availability

Not applicable.

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
