# Peer review of "Pathophysiological Features of Nigral Dopaminergic Neurons in Animal Models of Parkinson’s Disease"

_ijms, 2022, doi:10.3390/ijms23094508_

Round 1

Reviewer 1 Report

In the manuscript authors summarize shared or peculiar features of neuronal dysfunction of the dopaminergic neurons in different PD animal models, in the attempt to design a picture of the functional modifications occurring in nigral dopaminergic neurons during disease progression, preceding their eventual death.

This manuscript is a clear and concise compilation of research papers on the abnormalities and molecular mechanisms of neural function in major animal models of Parkinson's disease. The authors cite an appropriate balance of their own work and that of other groups.

The authors have paid much attention to defining what functional alteration occurs in the dopaminergic neuron population, preceding their final degeneration, due to altered expression or activity of specific ion channels or of the excitatory/inhibitory balance in the network.

First, the authors describe the neurotoxin-based models of Parkinson’s disease.

The effects of the 6-OHDA, rotenone, paraquat, beta-N-methyl amino-L-alanine and MPRP/MPP+ on the electrophysiological properties of dopaminergic neurons and molecular mechanisms of these toxin on alteration of dopaminergic neurons were summarized.

Second, the authors summarize the alpha-synuclein (Snca)-dependent effects onto functional properties dopaminergic neurons in substantia nigra pars compacta using Snca-expressing transgenic mice.

Third, the authors summarize the principal findings regarding genetic manipulation effects onto dopaminergic neurons in substantia nigra pars compacta using PINK1, Parkin, LRRK2, DJ-1, MitoPark and Ndufs of mitochondrial complex 1 gene-modified mice,

In conclusion, this manuscript would draw much attentions towards the inclusion of dopaminergic neurons excitability as an important parameter in studies focused on the cellular mechanisms of PD using animal models in vivo and in vitro.

Author Response

RE: We are thankful to Reviewer for his/her comments on our manuscript.

Reviewer 2 Report

I have some comments on this manuscript.

  1. Include some key components in the abstract section like any signaling pathways etc.
  2. In addition, write the limitations of your review at the end of the abstract section.
  3. Cite PMID: 34970114, PMID: 33510062, PMID: 33194526, and PMID: 31996329 for your sentence “Parkinson’s disease (PD) is the second most common neurodegenerative disease in humans characterized by a progressive demise of dopaminergic (DAergic) neurons of the substantia nigra pars compacta (SNpc)”.
  4. Elaborate the introduction section by discussing the role of some herbal plants like Mucuna pruriens, Withania somnifera, Tinospora cordifolia, and their bioactive components like ursolic acid and chlorogenic acid in the MPTP intoxicated mouse model.
  5. Discuss the role of vital transcription factor-like NF-kB, TFEB, and Nrf2 in different toxin-induced PD models.
  6. What are the other areas behind SNpc and ST that are affected in PD.?
  7. Concise the conclusion section and include a summary or discussion section just before this.
  8. 1 signaling figure will be needed to support the idea of your manuscript. Include both upstream and downstream regulators in this and show the effect of different transcription factor-mediated neuroprotection in this.
  9. Complete editorial checking will be needed to correct the grammatical and punctuation mistakes.

Author Response

  1. Include some key components in the abstract section like any signaling pathways etc.

 RE: According to Reviewer’s suggestion, we have added the following sentence in the Abstract: ‘The degeneration of nigral dopaminergic neurons is considered the hallmark of Parkinson’s disease (PD) and it is triggered by different factors, including mitochondrial dysfunction, Lewy bodies accumulation, neuroinflammation, excitotoxicity and metal accumulation'.

2. In addition, write the limitations of your review at the end of the abstract section.

RE: According to Reviewer suggestion, we added the following sentence in the last paragraph of the Abstract ‘We focus on dopamine neuron excitability and functional properties and summarize…’.

  1. Cite PMID: 34970114, PMID: 33510062, PMID: 33194526, and PMID: 31996329 for your sentence “Parkinson’s disease (PD) is the second most common neurodegenerative disease in humans characterized by a progressive demise of dopaminergic (DAergic) neurons of the substantia nigra pars compacta (SNpc)”.

RE: The indicated manuscripts suggested by Reviewer mainly deal with therapeutic strategies based on natural compounds with known anti-PD potential. For this reason, we believe that they would not be pertinent citations for the particular sentence mentioned by the Reviewer, which rather refers to a general concept of PD incidence in humans and to selective degeneration of midbrain dopaminergic neurons.

However, we have inserted the citation PMID: 33194526 at line 80 of the manuscript text, in the ‘Toxin-based PD Models’ section, as it is pertinent for our statement ‘Some neurotoxins have been shown to cause DAergic neurons degeneration with a high degree of selectivity, as they enter the neurons via dopamine transporter (DAT)’, and PMID31996329 in line 82, as it represents a recent Review on PD models.

  1. Elaborate the introduction section by discussing the role of some herbal plants like Mucuna pruriens, Withania somnifera, Tinospora cordifolia, and their bioactive components like ursolic acid and chlorogenic acid in the MPTP intoxicated mouse model.

RE: We are thankful to Reviewer for his/her comment. The beneficial effect of herbal plants and bioactive components in PD is a very interesting Research area. From our prospective, it will be interesting to investigate the effects of these compounds on excitability of dopaminergic neurons in PD models in the near future. However, with regard to the present manuscript, we do not include discussion on the role of beneficial therapies, but we rather focus on excitability modifications occurring in dopaminergic neurons of different PD models, at different stages of PD progression, as clearly stated in the Abstract (see our response to #2).

  1. Discuss the role of vital transcription factor-like NF-kB, TFEB, and Nrf2 in different toxin-induced PD models.

RE: We are thankful to Reviewer for his/her suggestion. As stated in our response to #4 for the role of beneficial therapeutics, we believe that discussing the role of vital transcription factors in toxin-induced PD models goes beyond the scope of the present manuscript.

  1. What are the other areas behind SNpc and ST that are affected in PD.?

RE: The severity of motor and non-motor symptoms may suggest that Parkinson’s disease patients would have a widespread and massive degeneration of neurons across the brain. On the contrary, it has been shown that the zones of pathology, up until the late stages of the disease, are rather discrete, affecting only small groups of neurons located in the brainstem. In addition, it has been reported that other localized regions are affected in PD, including the dopaminergic neurons of the adjacent ventral tegmental area and retrorubral field of the midbrain and those of the olfactory bulb, the noradrenergic neurons of the locus coeruleus, the cholinergic neurons of the pedunculopontine tegmental nucleus, the serotonergic neurons of the raphe nuclei, together with neurons of the dorsal motor nucleus of the vagus nerve. At much later stages of the disease, there is also some loss of neurons across the cortex (Halliday et al 1990, Mavridis et al 1991, Braak et al 2003, Zarow et al 2003, Langston 2006, Del Tredici and Braak 2013, Brettschneider et al 2015, French and Muthusamy 2018).

  1. Concise the conclusion section and include a summary or discussion section just before this.

RE: We are thankful to Reviewer for his/her suggestion. We have inserted a summary paragraph before the Conclusion section. The Conclusion section has been shortened accordingly. We also added a diagram (Figure 1) in this section (see response to Point #8).

  1. 1 signaling figure will be needed to support the idea of your manuscript. Include both upstream and downstream regulators in this and show the effect of different transcription factor-mediated neuroprotection in this.

RE: We are grateful to the Reviewer for pointing out the need of inserting a summarizing diagram. According to Reviewer suggestion, we have included a figure displaying the main membrane channels/receptors/transporters contributing to the changes of dopamine neuron excitability in the PD models mentioned in our manuscript.

  1. Complete editorial checking will be needed to correct the grammatical and punctuation mistakes.

 RE: We thank the Reviewer for his/her comment. We carefully checked out the text and corrected grammatical and punctuation mistakes.

Round 2

Reviewer 2 Report

Manuscript is revised as per my suggestion.